# Cystathionine γ-Lyase Self-Inactivates by Polysulfidation during Cystine Metabolism

**DOI:** 10.3390/ijms24129982

**Published:** 2023-06-10

**Authors:** Shoma Araki, Tsuyoshi Takata, Katsuhiko Ono, Tomohiro Sawa, Shingo Kasamatsu, Hideshi Ihara, Yoshito Kumagai, Takaaki Akaike, Yasuo Watanabe, Yukihiro Tsuchiya

**Affiliations:** 1Department of Pharmacology, Showa Pharmaceutical University, Machida 194-8543, Japan; araki@ac.shoyaku.ac.jp (S.A.); ttakata@nd.edu (T.T.); 2South Bend Campus, Indiana University School of Medicine, South Bend, IN 46617, USA; 3Department of Microbiology, Graduate School of Medical Sciences, Kumamoto University, Kumamoto 860-8556, Japan; onokat@kumamoto-u.ac.jp (K.O.); sawat@kumamoto-u.ac.jp (T.S.); 4Department of Biological Chemistry, Graduate School of Science, Osaka Metropolitan University, Sakai 599-8531, Japan; kasamatsu@omu.ac.jp (S.K.); iharah@omu.ac.jp (H.I.); 5Graduate School of Pharmaceutical Sciences, Kyusyu University, Fukuoka 812-8582, Japan; kumagai.yoshito.864@m.kyushu-u.ac.jp; 6Department of Environmental Medicine and Molecular Toxicology, Graduate School of Medicine, Tohoku University, Sendai 980-8575, Japan; takaike@med.tohoku.ac.jp

**Keywords:** cystathionine γ-lyase (CSE), polysulfidation, cysteine persulfide (Cys-SSH), redox regulation

## Abstract

Cystathionine γ-lyase (CSE) is an enzyme responsible for the biosynthesis of cysteine from cystathionine in the final step of the transsulfuration pathway. It also has β-lyase activity toward cystine, generating cysteine persulfide (Cys-SSH). The chemical reactivity of Cys-SSH is thought to be involved in the catalytic activity of particular proteins via protein polysulfidation, the formation of -S-(S)n-H on their reactive cysteine residues. The Cys136/171 residues of CSE have been proposed to be redox-sensitive residues. Herein, we investigated whether CSE polysulfidation occurs at Cys136/171 during cystine metabolism. Transfection of wild-type CSE into COS-7 cells resulted in increased intracellular Cys-SSH production, which was significantly increased when Cys136Val or Cys136/171Val CSE mutants were transfected, instead of the wild-type enzyme. A biotin-polyethylene glycol-conjugated maleimide capture assay revealed that CSE polysulfidation occurs at Cys136 during cystine metabolism. In vitro incubation of CSE with CSE-enzymatically synthesized Cys-SSH resulted in the inhibition of Cys-SSH production. In contrast, the mutant CSEs (Cys136Val and Cys136/171Val) proved resistant to inhibition. The Cys-SSH-producing CSE activity of Cys136/171Val CSE was higher than that of the wild-type enzyme. Meanwhile, the cysteine-producing CSE activity of this mutant was equivalent to that of the wild-type enzyme. It is assumed that Cys-SSH-producing CSE activity could be auto-inactivated via the polysulfidation of the enzyme during cystine metabolism. Thus, the polysulfidation of CSE at the Cys136 residue may be an integral feature of cystine metabolism, which functions to down-regulate Cys-SSH synthesis by the enzyme.

## 1. Introduction

Pyridoxal 5′-phosphate (PLP)-dependent enzymes, cystathionine β-synthase (CBS), and cystathionine γ-lyase (CSE) are key constituents of the transsulfuration pathway, in that CSE catalyzes cystathionine generated by CBS to generate cysteine [1]. In brief, CBS catalyzes the condensation of serine and homocysteine to cystathionine, which again is cleaved to cysteine, ammonia, and 2-oxobutyrate, which are catalyzed by CSE. Thus, the sulfur is transferred from homocysteine to cysteine via cystathionine. CSE also plays an important role in the production of cysteine persulfide (Cys-SSH) from cystine by its β-lyase activity [2]. Cys-SSH and related species, such as glutathione persulfide, have been suggested to act as powerful antioxidants and cellular protectants and may serve as reduction/oxidation (redox) signaling intermediates [3]. It was recently shown that cysteinyl-tRNA synthetase (CARS) is a new Cys-SSH synthase, in that CARS utilizes cysteine generated by CBS/CSE [4]. Additionally, CSE may still play a major role in the Cys-SSH generation from cystine as the substrate, especially under pathological conditions associated with oxidative and electrophilic stress, where intracellular cystine concentrations considerably approach the high Km value of CSE. Unlike CBS, which is constitutively expressed, CSE is transcriptionally inducible by various stimuli, such as endoplasmic reticulum stress and dietary restriction [5,6].

Both CBS and CSE undergo several post-translational modifications that can alter their enzymatic activity or sub-cellular localizations. CBS is a hemeprotein, and both carbon monoxide (CO) and nitric oxide (NO) bind to ferrous heme and inhibit the enzyme, promoting hydrogen sulfide (H_2_S) synthesis from cysteine by CSE [7]. CSE is not a hemeprotein, but endogenous *S*-nitrosylated CSE at Cys136/171 has been observed in the mouse liver [8]. Recently, it was shown that the physiological NO donor *S*-nitrosoglutathione (GSNO) inhibits human CSE H_2_S-generating activity from cysteine via its *S*-nitrosylation at Cys137 (equivalent to Cys136 in mouse and rat) [9]. Nitration and polysulfidation modification of CSE have been also reported, but the significance of the site-specific modification is not fully explored [10,11]. Notably, the chemical reactivity of Cys-SSH has been thought to regulate the catalytic activity of particular proteins by their cysteine polysulfidation.

Thus, our current aim was to determine whether cystine-derived Cys-SSH would modify cysteine polysulfidation on CSE during enzyme catalysis. Experiments were also performed to determine whether this modification was related to the regulation of Cys-SSH synthesis, including the effect of the exogenous polysulfide donor Na_2_Sn (n = 2, 3, 4) on CSE function. We propose that polysulfidation at Cys136/171 is a fundamental part of the enzyme’s normal cystine metabolism and functions to regulate Cys-SSH synthesis by CSE.

## 2. Results

### 2.1. Cys136 and/or Cys171 Are Redox Sensors of CSE during the β-Lyase Activity toward Cystine to Generate Cys-SSH

Endogenous CSE was reported to be *S*-nitrosylated at Cys171 and Cys136/Cys171 in the wild-type mouse kidney and liver, respectively [8]. The Cys137 residue in humans (equivalent to Cys136 in mice and rats) was identified as most significantly contributing to the *S*-nitrosoglutathione-mediated CSE inhibition in H_2_S synthesis from cysteine [9]. Therefore, we first generated a rat CSE, in which Val was substituted for Cys136 (C136V), Cys171 (C171V), or Cys136/171 (C136V/C171V), and it was characterized by its β-lyase activity using β-chloro-L-alanine (β-CA) (Figure 1) [12] or cystine (Figure 2) as substrates.

We constructed and purified rat CSE by site-specific separation of the GST tag from proteins expressed using pGEX-6P vectors. All CSEs were at least 90% pure and gave a major band at ∼45 kDa on the SDS-PAGE with Coomassie Brilliant Blue staining (Figure 1).

The enzyme activities of all recombinant CSEs were similar when β-CA was employed as a substrate (Table 1). However, the β-lyase reactions of cystine to generate Cys-SSH in mutants, especially C136V and C136V/C171V, were larger than that of the wild-type, as determined with a sulfane sulfur-specific fluorescent probe, SSP4 (Table 2). As shown in Appendix A, the kinetic curves of these two substrates and different mutants are quite different, in that C136V and C136V/C171V mutants showed larger cystine-, but not β-CA-, metabolic activity.

We recently reported that fluorescence analyses of SSP4, which determine its high selectivity and sensitivity to sulfane sulfurs, even with the interfering presence of other species, such as amino acids and metal ions [13]. We also examined the β-lyase activity toward cystine to generate Cys-SSH in COS-7 cells, expressing wild-type, C136V, C171V, or C136V/C171V CSE. There was an equal amount of CSE in cells over-expressing each CSE examined by Western blotting using the anti-CSE antibody (Figure 2A). Endogenous CSE could not be detected because of its negligible expression compared to that of overexpressed CSEs. There was a similar SSP4 fluorescence in wild-type or C171V CSE-expressing cells. Meanwhile, there was a significant increase in SSP4 fluorescence in C136V or C136V/C171V CSE-expressing cells, when compared with wild-type CSE-expressing cells (Figure 2B,C). On the other hand, the γ-lyase activity toward cystathionine to generate cysteine did not change in C136V/C171V, as compared to wild-type CSE (Figure 3). These results demonstrate that the downregulation of CSE selectively occurred during the β-lyase step to synthesize Cys-SSH from cystine via Cys136 and/or Cys171 modifications.

### 2.2. Exogeneous Cys-SSH and Polysulfides Inhibit CSE β-Lyase Activity toward Cystine

The above data suggest the causal role of CSE-derived Cys-SSH or its derivatives that lead to consequent self-dysfunction. To test this concept further, we investigated whether exogenous polysulfide donors Na_2_Sn (n = 2–4) could inhibit CSE β-lyase activity and, if so, posttreatment with reducing agent DTT could restore the activity. Pretreatment of immobilized CSE with 10 μM Na_2_S_4_ caused the inhibition of its β-lyase activity toward cystine, which was completely recovered by the addition of DTT (Figure 4A). We observed that Tris (2-carboxyethyl) phosphine (TCEP) could also recover the 10 μM Na_2_S_4_-induced CSE β-lyase activity toward cystathionine to generate Cys-SSH (data not shown). We also determined the effect of sulfur contents of polysulfide donors on this inhibition. All the polysulfide species inhibited the β-lyase activity of CSE with approximate IC_50_ values of 5 μM for Na_2_S_4_, 7 μM for Na_2_S_3_, and 20 μM for Na_2_S_2_ (Figure 4B). We also determine the effect of CSE-derived Cys-SSH from cystine on the β-lyase activity of CSE. We incubated cystine with buffer alone, or recombinant CSE, and then examined the Cys-SSH level with SSP4. There was a significant increase in SSP4 fluorescence when cystine was incubated with CSE, but not buffer alone (Figure 4C). Next, we assessed whether this enzymatically synthesized Cys-SSH may be involved in the inhibition of CSE activity. As shown in Figure 4D, cystine alone had minimal effects on the β-lyase activity of CSE. However, when immobilized, CSE was incubated with the products resulting from the CSE/cystine, and there was a significant effect on inhibition. Thus, not only CSE-derived Cys-SSH, but also various sulfur species, including persulfides and polysulfides, may inhibit the β-lyase activity of CSE.

### 2.3. Cys136 Is an Essential Site for the Inactivation of CSE by Polysulfides

Based on the above observations, CSE is responsible for the target of exogeneous polysulfides. We further clarified whether Cys136 and/or Cys171 were crucial sites for inhibition by exogeneous polysulfides regarding CSE activity. Treatment of the immobilized CSE with 10 μM Na_2_S_4_ resulted in the inhibition of enzyme activity. C171V mutant was also inhibited by Na_2_S_4_ as to wild-type CSE. However, C136V and C136V/C171V mutants appeared to be partially resistant to Na_2_S_4_-induced inactivation (Figure 5A). Next, we also determined the effects of CSE-derived Cys-SSH on the CSE activity. With CSE-derived Cys-SSH, no decrease in activity was noted with C136V and C136V/C171V mutants, whereas the wild-type and C171V mutants were significantly inactivated (Figure 5B). Cys-SSH may be rather specific for Cys136 residue modification, leading to a decrease in enzyme activity. Meanwhile, Na_2_S_4_ may target other Cys residues, besides Cys136, to inhibit activity.

### 2.4. Cys136 and/or Cys 171 Represents the Sites of Polysulfidation on CSE

The above experiments clearly show the susceptibility of Cys136 residues in CSE to persulfides and polysulfides. We developed a biotin-polyethylene glycol-conjugated maleimide (biotin-PEG-MAL) capture method, by which the polysulfidated proteins can be eluted by adsorbing biotin-PEG-MAL-modified proteins to immobilized avidin resin and cleaving polysulfides with a reducing agent [4,14]. We examined whether Cys136 and/or Cys171 residues in CSE are accessible to polysulfidation during catalysis using a biotin-PEG-MAL capture assay. Using densitometric analysis of CSE polysulfidation in a biotin-PEG-MAL capture assay and aliquots of CSEs that had not been subjected to the assay (“Total CSE”), we observed that incubation with cystine led to an increase either in wild-type, C136V, or C171V CSE polysulfidation. Meanwhile, these increases were not observed in the C136V/C171V mutants, indicating that simultaneous polysulfidation of Cys136 and Cys171 occurred and that individual polysulfidation of either residue was not detected in the assay conditions employed in the study (Figure 6).

## 3. Discussion

In the present study, we identified a novel downregulation of CSE via its polysulfidation, both during cystine metabolism and by exogenous polysulfides. CSE produces Cys-SSH from cystine by its β-lyase activity in the condition where the intracellular cystine concentration is considerably approaching the high Km value of CSE under pathological conditions, such as oxidative and electrophilic stress. Electrophilic stress is preferably defined as the overwhelming production of reactive electrophilic and oxygen species that exceeds the ability of anti-chemoprotective and anti-oxidant systems to neutralize both types of reactive species [15]. In such an intracellular environment, the self-regulatory mechanism of CSE β-lyase activity toward cystine may be regulated. In this study, Na_2_Sn (n = 2–4) was used as an exogenous polysulfide donor. Polysulfides with high sulfur contents showed stronger inhibitory effects on CSE activity (Figure 4B), consistent with a recent report that Ca^2+^/calmodulin-dependent protein kinase (CaMK) II and CaMKIV are inhibited in the same way [16,17]. As noted earlier, it has been proposed that increasing the number of sulfur atoms would lead to enhanced thiol reactivity due to increased nucleophilicity [18]. GSSH concentrations in tissues are >100 μM in the brain and about 50 μM in other major organs, including the heart and liver in the physiological condition [3], thereby acting as major donors for protein polysulfidation. Considering the IC_50_ values of ≈20 μM for Na_2_S_2_ (Figure 4B), it may be assumed that basal polysulfidated CSE in the present study was most probably a consequence of GSSH function in cells. Meanwhile, our results showed that the polysulfidation and inhibition of CSE are reversed by post-addition of DTT (Figure 4A). In a cellular context, protein polysulfidation is reversed by the action of thioredoxin (Trx)/Trx reductase (TrxR) [19]; thus, CSE could be reversibly regulated by polysulfides and the Trx/TrxR system. A previous report indicated that the Cys residues with lower pKa values, which depend upon their local charge environment, exist as thiolate anions under normal conditions and are susceptible to polysulfidation [20]. However, as of today, no consensus motif is known to fully predict the polysulfidation sites in the modified protein.

There are several reports, including from our group, that the enzymes were modified via both *S*-nitrosylation and polysulfidation at the same Cys residues. Among them, in the anti-apoptotic actions of NF-κB in response to TNF-α activation, NF-κB transcriptional activity in response to TNF-α is initially activated by its p65 polysulfidation and then inhibited by its *S*-nitrosylation, sequentially, at the same Cys residue [21]. Similarly, polysulfidation of endothelial NO synthase (eNOS) decreases eNOS *S*-nitrosylation at the same Cys residue and increases eNOS phosphorylation and dimer stability, resulting in increased NO bioavailability [22]. We reported previously that *S*-nitrosylation and polysulfidation of CaMKII at Cys6 abolish its catalytic activity in an ATP competitive fashion [16,23]. Endogenous *S*-nitrosylated CSE at Cys136/171 or at Cys171 has been observed in wild-type mouse livers or kidneys, respectively [8]. The physiological NO donor GSNO inhibits CSE H_2_S-generating activity from cysteine via its *S*-nitrosylation at human Cys137 (equivalent to Cys136 in the mouse and rat) [9]. Thus, *S*-nitrosylation of and polysulfidation of CSE at Cys136 inhibit its activity when cysteine and cystine are metabolized, respectively. Notably, the latter occurs not only by exogenous polysulfides, but also during catalysis.

In principle, Cys residues were substituted by Ser or Ala; the activity of C136S/C171S mutant CSE was decreased compared to that of the wild-type enzyme (data not shown). We generated the CSE mutant, C136V, C171V, and C136V/C171V (in which Cys residues were replaced with Val), since the structural difference between valine and alanine/serine may be important in the conformation of the mutated protein [17]. In fact, a previous report showed that human Cys137 (equivalent to Cys136 in the mouse and rat) mutant was functionally impaired in its H_2_S generating activity from cysteine compared to wild-type CSE [9]. This also suggests that CSE-derived H_2_S from cysteine could not inhibit the enzyme activity via Cys137. The impact of the Cys residues, including Cys136 on the CSE structure and function, have been extensively discussed [9]. Polysulfidation occurred in both Cys136 and Cys171 residues in CSE (Figure 6) and Cys136, but not in Cys171, which is the representative polysulfidated residue for the inhibition of CSE enzyme activity (Figure 2 and Figure 5B). Rat liver CSE contains twelve cysteine residues. To determine which cysteine residue(s) are sensitive to self-inactivation during cystine metabolism, we generated mutants of rat CSE in which each of the twelve cysteine residues were replaced by valine residues. We constructed and purified rat CSE by site-specific separation of the GST tag from proteins expressed using pGEX-6P vectors. All CSEs were at least 90% pure and gave a major band at ∼45 kDa on the SDS-PAGE with Coomassie Brilliant Blue staining (Appendix A). The C136V mutant showed larger β-lyase activity toward cystine than that of the wild-type CSE (Appendix A). Thus, Cys136 is an essential site for the self-inactivation of CSE during cystine metabolism. The apparent inhibition, observed with varying concentrations of cystine (50–1000 μM), of wild-type CSE compared with C136V, decreased in both the substrate Km and Vmax to 49% and 30%, respectively (Table 2). This indicates that the CSE inhibition mechanism during its β-lyase catalysis toward cystine tended to be uncompetitive with cystine. That is, as an uncompetitive inhibitor, CSE-derived Cys-SSH may only bind to the CSE-cystine complex at Cys136 dominantly. Cys136 modification may occur within the CSE holoenzyme, but exogenous polysulfides also inhibited CSE β-lyase activity (Figure 5). However, this study does not eliminate the possibility of intra-subunit modification within the homotetrameric holoenzyme in CSE.

In summary, earlier studies showed that neuronal NOS self-inactivated by heme-NO complex forms during arginine metabolism [24]. Although, whether CSE-derived H_2_S from cysteine could inhibit its enzyme activity is still unknown, to the best of our knowledge, and the present observation is the first report to show that the enzyme in the transsulfuration pathway is self-regulated during catalysis. We recently showed that CSE and inducible NOS were up-regulated in macrophage cells in response to LPS-induced inflammation [25]. We are initiating a study to determine whether inducible NOS-derived NO inactivates the CSE β-lyase activity toward cystine and, if so, which types of modification of its reactive cysteine residue are involved. It was recently reported that excess intracellular Cys-SSH was exported into the extracellular space by a cystine-dependent transporter to maintain intracellular redox homeostasis [26]. It is assumed that self-regulation of CSE and cysteine persulfide antiporter work in parallel to prevent the accumulation of surplus Cys-SSH in the cells.

## 4. Materials and Methods

### 4.1. Materials

The cDNA for rat liver CSE was a generous gift from Dr. Nozomu Nishi [27] (Life Science Research Center, Kagawa University, Kagawa, Japan). The anti-CSE antibody was prepared, as described previously [28]. Sulfane sulfur probe 4 (SSP4) and Na_2_Sn (n = 2–4), biotin-HPDP, a water-soluble biotin-labeling reagent, 4,4-Dithiopyridine, and N,N’-Bis(2-hydroxy-3-sulfopropyl)tolidine were obtained from Dojindo laboratories (Kumamoto, Japan). Neutravidin-agarose was obtained from Thermo Fisher Scientific (Waltham, MA, USA). Pyruvate oxidase, peroxidase, thiamine pyrophosphate, and β-chloro-L-alanine (β-CA) were obtained from Sigma-Aldrich (St Louis, MO, USA). All other materials and reagents were of the highest quality available from commercial suppliers.

### 4.2. Plasmid Construction

The rat CSE mutants C136V, 171V, or C136V/C171V (i.e., a mutant bearing replacement of both Cys136 and Cys171 with Val residues) were subcloned into a pME18S vector. The nucleotide sequences of each mutant were confirmed by DNA sequencing.

### 4.3. CSE Purification

Recombinant rat CSE was expressed in *E. coli* (DH5α) using pGEX-6P and purified by using GSH accept (Nacalai Tesque, Kyoto, Japan). Protein concentrations were determined by the Bradford method, using BSA as the protein standard.

### 4.4. Measurement of CSE Activity

The Cys-SSH produced by CSE from cystine as a substrate was measured by using SSP4. We initially performed an enzyme titration curve and a time course for SSP4 assay. The experiment was performed in the reaction with substrate concentration at 1 mM of cystine. The velocity is obtained from the slope of the linear part of the curve. Thus, enzyme concentration at 50 μg/mL and the assay time for 20 min were selected for the assay. Recombinant CSEs (50 μg/mL) were incubated in 20 mM HEPES (pH 7.5), containing 50 μM PLP with buffer alone or 1 mM cystine for 20 min at 37 °C. Samples of interest were reacted with 10 μM SSP4 in 20 mM Tris-HCl (pH 7.4) in the presence of 1 mM cetyltrimethylammonium bromide in the dark for 10 min at room temperature. Fluorescence intensities of the resultant solutions were determined using a microplate reader (BioTek Synergy HTX Multimode Reader, Agilent Technologies, Inc., Santa Clara, CA, U.S.A.) with an excitation wavelength of 485/20 nm and an emission wavelength of 528/20 nm. In some studies, it was necessary to remove the polysulfide donor which reacts with SSP4 and reducing agent, which decomposes Cys-SSH produced by CSE to cysteine. Thus, we established an immobilized CSE assay. GST-CSE was expressed for use in *E. coli* (DH5α) using pGEX-6P, which was subsequently immobilized with GSH-agarose. Treatment of Cys-SSH and Na_2_Sn (n = 2–4) and reducing agent were removed by centrifugation, and immobilized CSE was washed with 50 mM HEPES (pH 7.5) three times followed by incubation with 50 µM PLP, as well as 1 mM cystine, at 37 °C for 20 min. The Cys-SSH contents in the samples of interest were analyzed, as described above.

Pyruvate produced by CSE from β-CA as a substrate was measured by coupling a color enzymatic reaction with pyruvate oxidase and peroxidase, as described previously [12], with a minor modification. Briefly, recombinant CSEs (13 μg/mL) were incubated in 100 mM Tris-phosphate (pH 8.0), containing 35 μM PLP with buffer alone or 15 mM β-CA 10 min at 37 °C. The reaction is terminated by the addition of 4,4-dithiopyridine to inactivate the enzyme, by the masking of free thiol groups on the active site. The pyruvate produced is oxidized by pyruvate oxidase in the presence of thiamine pyrophosphate and divalent magnesium to liberate hydrogen peroxide. A N,N’-Bis(2-hydroxy-3-sulfopropyl)tolidine is oxidized by hydrogen peroxide with peroxidase to produce the colored matter with an absorption maximum at 674 nm.

Cysteine produced by CSE from cystathionine was measured by DTNB assay [29]. Briefly, purified CSEs (30 μg/mL) were incubated with 1 mM cystathionine in 40 mM borate buffer (pH 8.2), containing 50 µM PLP and 1 mM DTNB at 30 °C for 10 min. The development in absorption at 412 nm due to the formation of the nitrobenzene thiolate anion was measured.

### 4.5. Live-Cell Fluorescence Imaging of Cys-SSH

COS-7 cells were maintained in Dulbecco’s modified Eagle medium (DMEM), containing 10% fetal calf serum (FBS) and 1% penicillin streptomycin in a humidified atmosphere at 37 °C. COS-7 cells were seeded in a 6 cm dish at a density of 4 × 10^5^ cells per dish. Transient transfection of COS-7 cells was performed with Lipofectamine 3000 (Thermo Fisher Scientific). Briefly, 2.5 µg of plasmid, 5 µL of P3000 reagent, and 7.5 µL of Lipofectamine 3000 reagent were mixed in Opti-MEM, and the mixture was added to the cells. After 24 h incubation, transfected COS-7 cells were re-seeded in eight-well chamber slides (µ-Slide 8 well high, ibidi GmbH, Gräfelfing, Germany) at a density of 2.4 × 10^4^ cells per well for imaging, as well as in a 6 cm dish at a density of 4 × 10^5^ cells per dish for Western blotting. After 48 h incubation, the cells were washed once with serum-free DMEM, followed by incubation with 20 µM SSP4 in serum-free DMEM, containing 500 µM cetyltrimethylammonium bromide at 37 °C for 15 min. After removing the excess probes from the cells and washing them with HBSS, fluorescence images were captured using a Nikon A1R+ confocal laser scanning microscope with an ECLIPSE Ti-E inverted microscope and a 20× objective lens. The excitation wavelength was 488 nm, and the emission wavelength was 500/50 nm. The fluorescence intensity of images was calculated by using NIS-Elements software (Nikon, Tokyo, Japan).

### 4.6. Detection of Polysulfidated CSE Using a Biotin-Polyethyleneglycol-Conjugated Maleimide Capture Assay

Polysulfidated proteins were detected using a biotin-polyethylene glycol-conjugated maleimide (biotin-PEG-MAL) capture assay [4,14]. Briefly, recombinant CSEs (100 μg/mL) were incubated in 30 mM HEPES (pH 7.5), containing 50 μM PLP with buffer alone or 1 mM cystine containing 1 mM biotin-PEG-MAL for 60 min at 37 °C. After biotinylated CSEs in the mixture were captured and enriched with Neutravidin-agarose (Thermo Fisher Scientific), polysulfidated CSEs were collected and subjected to Western blotting for CSE.

### 4.7. Statistical Analysis

All results are represented as the mean ± SE of at least three determinations. The statistical evaluation was performed using a one-way ANOVA test. We considered *p* < 0.05 to be statistically significant.

## Data Availability

Not applicable.

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
