# Peer review of "Cystathionine γ-Lyase Self-Inactivates by Polysulfidation during Cystine Metabolism"

_ijms, 2023, doi:10.3390/ijms24129982_

Round 1

Reviewer 1 Report

Cystathionine γ-lyase (CSE) catalyzes the PLP-dependent cleavage of cystathionine to cysteine, ammonia and 2-oxobutyrate. It also acts as β-lyase acting on cystine yielding cysteine persulfide (Cys-SSH), ammonia and pyruvate. The authors showed that during the reaction with cystine the product Cys-SSH polysulfidated cysteine-136 of CSE whereby the enzyme became inhibited. Thus cystine inactivates the β-lyase activity, whereby the γ-lyase activity is not affected. The manuscript is of interest but the description of introduction, results and discussion must be improved.

Minor remarks

Line 44: “CSE catalyzes cystathionine generated by CBS to generate cysteine”. Do you mean: CBS catalyzes the condensation of serine and methionine to cystathionine which again is cleaved to cysteine, ammonia and 2-oxobutyrate catalyzed by CSE. Thus the sulfur is transferred form methionine to serine yielding cysteine?

Line 163 (B)? The sentence belongs to Fig. 4A. The legend to Fig. 4B is missing.

Line 219: what is “electrophilic stress”?

The English language is of reasonable quality but on some places the wording needs to be improved.

Author Response

Response to Reviewer 1 Comments

The manuscript is of interest but the description of introduction, results and discussion must be improved.

Thank you for your positive comments to our manuscript.

Point 1:Line 44: “CSE catalyzes cystathionine generated by CBS to generate cysteine”. Do you mean: CBS catalyzes the condensation of serine and methionine to cystathionine which again is cleaved to cysteine, ammonia and 2-oxobutyrate catalyzed by CSE. Thus the sulfur is transferred form methionine to serine yielding cysteine? 

Response 1:In brief, CBS catalyzes the condensation of serine and homocysteine to cystathionine which again is cleaved to cysteine, ammonia and 2-oxobutyrate catalyzed by CSE. Thus, the sulfur is transferred from homocysteine to cysteine via cystathionine. We made a statement in this regard (lanes 44-47, highlighted in red).

Point 2:Line 163 (B)? The sentence belongs to Fig. 4A. The legend to Fig. 4B is missing.

Response 2:We appreciate the reviewer for pointing out this mistake. We added the legend to figure 4B (lanes 179-187, highlighted in red).

Point 3:Line 219: what is “electrophilic stress”?

Response 3:Electrophilic stress appears to be preferably defined as the overwhelming production of reactive electrophilic and oxygen species that exceeds the ability of anti-chemoprotective and anti-oxidant systems to neutralize both types of reactive species (High Throughput. 2018 Apr 27;7(2):12. doi: 10.3390/ht7020012.). We made a statement in this regard (lanes 240-243, highlighted in red).

Reviewer 2 Report

The authors have done a good job. The manuscript clearly shows that the polysulfidation of CSE at Cys136 is important for cystine metabolism.

There are some minor comments to make the manuscript better :

1. in line 120- it says that the y-lyase activity decreased a little in C136/C171V as compared to wild-type CSE, however from the figure it is clear that it is ns. How can you claim that ?

2. Figure 2A, please add the protein ladder and also where is the endogenous CSE in the blot ?

3. In figure 2C, please add ns for WT versus C171V. 

4. In figure 4A, can you show all the individual points for the replicates to better convince the readers. 

5. In figure 4C, it is very confusing with the inset and the labelling. Please make it more clear by either having 2 panels for it or better labelling. 

6. Line 177 says that C171V mutant demonstrated no remarkable effects. It would be better if you can also make it very clear that it looked similar to WT CSE. 

7. The original blots should also have protein ladder included in all the figures. 

Author Response

Response to Reviewer 2 Comments

The authors have done a good job. The manuscript clearly shows that the polysulfidation of CSE at Cys136 is important for cystine metabolism.

Thank you for your positive comments to our manuscript.

Point 1: in line 120- it says that the γ-lyase activity decreased a little in C136/C171V as compared to wild-type CSE, however from the figure it is clear that it is ns. How can you claim that ?

Response 1: Based on the reviewer’s comments, we have rephrased as followed. On the other hand, the γ‐lyase activity toward cystathionine to generate cysteine did not change in C136V/C171V as compared to wild-type CSE (Figure 3) (lane 131, highlighted in red).

Point 2: Figure 2A, please add the protein ladder and also where is the endogenous CSE in the blot ?

Response 2: Based on the reviewer’s comments, we have added the protein ladder in figure 2A. Endogenous CSE was expressed in COS-7 cells (see attached figure 1) but it could not detected because of its negligible expression compared to that of overexpression CSEs. We made a statement in this regard (lanes 125-127, highlighted in red).

Point 3: In figure 2C, please add ns for WT versus C171V.

Response 3: Based on the reviewer’s comments, we have added ns for WT versus C171V in figure 2C (see figure 2).

Point 4: In figure 4A, can you show all the individual points for the replicates to better convince the readers. 

Response 4: We did not test whether DTT could recover polysulfide donors-induced inhibition of CSE β-lyase activity toward cystine in each point. But, we observed Tris(2-carboxyethyl)phosphine (TCEP) could also recover the 10 μM Na2S4-induced CSE β-lyase activity toward cystathionine to generate Cys-SSH (see attached figure 2). We would like to show the data in the concentration of 10 μM Na2S4 in the present study and we hope readers could convince that CSE is reversibly inactivated by Na2S4. We made a statement in this regard (lanes 156-159, highlighted in red).

Point 5:  In figure 4C, it is very confusing with the inset and the labelling. Please make it more clear by either having 2 panels for it or better labelling.

Response 5: Based on the reviewer’s comments, we have remade the figure 4C and 4D (see figure 4). We made a statement in this regard (lanes 181-187, highlighted in red).

Point 6:  Line 177 says that C171V mutant demonstrated no remarkable effects. It would be better if you can also make it very clear that it looked similar to WT CSE.

Response 6: Based on the reviewer’s comments, we have rephrased that C171V mutant was also inhibited by Na2S4 as to wild-type CSE (lanes 194-195, highlighted in red). 

Point 7:  The original blots should also have protein ladder included in all the figures.

Response 7: Based on the reviewer’s comments, we have added the protein ladder in original blots.

Reviewer 3 Report

Authors describe the self-inactivation mechanism of CSE by polysulfide modifications. Authors use  SSP4 fluorescence assay to perform the quantification of sulfane sulfur imaging and quantification and identified Cys136 may be the essential residue for regulation. However, I have some concerns regarding the acceptance of the manuscript.

Major issue:

1.       Authors have been using SSP4 for labeling the polysufide species. However, There have been reports about the misinterpretation of polysulfide speciation using alkylating reagents such maleimide, iodoacetamide based reagents. (British Journal of Pharmacology (2019) 176 646–670).How can authors justify the reliability of the quantification for polsylufide species from the CSE reaction?

2.       Figure 6: There is ~ 3 fold difference between without and with cystine treatment in C136V mutant. Why does the statistical analysis show n.s.? Besides, in C136V mutant, from the visual observation of the gel imaging, without cystine treatment, there is no detectable amount of polysulfidated CSE but there is detectable amount of polysulfidated CSE after AC. The signal difference should be much more than ~3 fold from the bar plot. Therefor the bar plot could be a misinterpretation of the WB.

3.       Figure 5 & Figure 6: There is disconnect between CSE beta-lyase activity and CSE polysulfidation. The CSE activity is significantly inhibited in WT CSE and resolved in C136V mutant. However, the CSE overall polysulfidation level between WT and C136V isn’t significant. Therefore, the conclusion of C136’s function in polysulfide inactivation is questionable

4.       Throughout the manuscript, authors only detect overall polysulfidation modification and maleimide-based labeling can misinterpret polysulfide speciation (British Journal of Pharmacology (2019) 176 646–670). There have been reports using mass spectrometry-based method to detect the polysulfide modification from enzymes (ACS Catal. 2020, 10, 16, 8981–8994). Authors should consider site-specific detection of polysulfides in CSE.

5.       Figure 5A: there is still significant activity difference in double mutant. CSE has 10 cysteine sites. Can author explain the function of other cysteine in polysulfide inactivation from the CSE structure point of view? Are those cysteine far from the active site or fully solvent exposed and why are they not considered for the mutagenesis studies in this manuscript?

Minor issue:

1.       Table 1 & Table 2: Authors should have the kinetic curves of these two substrates and different mutants in supplementary figures so that others can check the fitting of kinetic curves.

2.       The method part has the pyruvate quantification by coupling assay. However, the main text doesn’t describe this method at all. It should be deleted.

Author Response

Response to Reviewer 3 Comments

Major issue:

Point 1: Authors have been using SSP4 for labeling the polysufide species. However, There have been reports about the misinterpretation of polysulfide speciation using alkylating reagents such maleimide, iodoacetamide based reagents. (British Journal of Pharmacology (2019) 176 646–670).How can authors justify the reliability of the quantification for polsylufide species from the CSE reaction? 

Response 1: As the reviewer indicated, we reported that the final alkylated mixture may not represent the speciation that prevailed before alkylation (British Journal of Pharmacology (2019) 176 646–670). But in terms of fluorescence analyses of SSP4, we recently reported that fluorescence analyses of SSP4 determines its high selectivity and sensitivity to sulfane sulfurs, even with the interfering presence of other species, such as amino acids and metal ions (Redox Biology 56 (2022) 102433).

We initially performed that enzyme titration curve (attached figure 3A) and time course (attached figure 3B) for SSP4 assay. The experiment was performed in the reaction with substrate concentration at 1 mM of cystine. The velocity is obtained from the slope of the linear part of the curve. Thus, enzyme concentration at 1 μg/20 μl: 50 μg/ ml in the text (panel A) and the assay time for 20 min (panel B) were selected for the assay (see attached figures). We made a statement in this regard (lanes 345-349, highlighted in red).

Point 2: Figure 6: There is ~ 3 fold difference between without and with cystine treatment in C136V mutant. Why does the statistical analysis show n.s.? Besides, in C136V mutant, from the visual observation of the gel imaging, without cystine treatment, there is no detectable amount of polysulfidated CSE but there is detectable amount of polysulfidated CSE after AC. The signal difference should be much more than ~3 fold from the bar plot. Therefor the bar plot could be a misinterpretation of the WB.

Response 2: Based on the reviewer’s comments, we did re-tested and re-evaluated the CysS-(S)n-H in CSEs during cystine metabolism in Fig. 6. The difference between without and with cystine treatment in C136V mutant was statistically significant but that in C136V/C171V mutant was not. Thus, simultaneous polysulfidation of Cys136 and Cys171 occurred and that individual polysulfidation of either residue was not detected in the assay conditions employed in the study. We would like to show the representative data in figure 6 in the present study, since the results are the mean ± SE of five-six independent experiments (see attached original blots). We made a statement in this regard (lanes 221-225, highlighted in red).

Point 3: Figure 5 & Figure 6: There is disconnect between CSE beta-lyase activity and CSE polysulfidation. The CSE activity is significantly inhibited in WT CSE and resolved in C136V mutant. However, the CSE overall polysulfidation level between WT and C136V isn’t significant. Therefore, the conclusion of C136’s function in polysulfide inactivation is questionable

Response 3: It is the legitimate concern. As the reviewer indicated, the treatment of CSE with CSE-derived Cys-SSH resulted in the similar inhibition of its β-lyase activity toward cystine in wild-type and C171V mutant. Meanwhile, no decrease in activity was noted with C136V and C136V/C171V mutants (Figure 5B). As we noted above in the response to #2 comment, the polysulfidation were occurred in both Cys136 and Cys171 residues in CSE, and Cys136 but not Cys171 is the representative polysulfidated residue for the inhibition of CSE enzyme activity. We made a statement in this regard (lanes 289-291, highlighted in red).

Point 4: Throughout the manuscript, authors only detect overall polysulfidation modification and maleimide-based labeling can misinterpret polysulfide speciation (British Journal of Pharmacology (2019) 176 646–670). There have been reports using mass spectrometry-based method to detect the polysulfide modification from enzymes (ACS Catal. 2020, 10, 16, 8981–8994). Authors should consider site-specific detection of polysulfides in CSE.

Response 4: We thank the reviewer for the valid point. As the reviewer indicated, we have reported that the final alkylated mixture using N-ethylmaleimide (NEM) or monobromobimane (MBB) may not represent the speciation that prevailed before alkylation (British Journal of Pharmacology (2019) 176 646–670). We would like to remind that we developed the biotin-PEG-MAL (BPM) capture method by which the polysulfidated proteins can be eluted by adsorbing BPM-modified proteins to immobilized avidin resin and cleaving polysulfides with a reducing agent (See an attached figure. 4 from Supplementary Fig. 3b. Nat Commun 2017, 8, (1), 1177). We made a statement in this regard (lanes 214-217, highlighted in red). Although mass spectrometry-based method to detect the polysulfide modification from enzymes is pretty valuable, we would like to show the BPM capture method using CSE mutants in the present study and we hope, in the future, to carry out further studies to clarify this question.

Point 5: Figure 5A: there is still significant activity difference in double mutant. CSE has 10 cysteine sites. Can author explain the function of other cysteine in polysulfide inactivation from the CSE structure point of view? Are those cysteine far from the active site or fully solvent exposed and why are they not considered for the mutagenesis studies in this manuscript?

Response 5: We thank again the reviewer for the valid point. Rat liver CSE contains twelve cysteine residues. To determine which cysteine residue(s) are sensitive to self-inactivation during cystine metabolism, we generated mutants of rat CSE in which each of the twelve cysteine residues were replaced by valine residues. We constructed and purified rat CSE by site-specific separation of the GST tag from proteins expressed using pGEX-6P vectors. All CSEs were at least 90% pure and gave a major band at ∼45 kDa on SDS–PAGE with Coomassie Brilliant Blue staining (Supplementary Figure 2A). The C136V mutant showed the larger β-lyase activity toward cystine than that of the wild-type CSE (Supplementary Figure 2B). Thus, Cys136 is an essential site for the self-inactivation of CSE during cystine metabolism. We made a statement in this regard in discussion section (lanes 291-300, highlighted in red).

 Minor issue:

Point 1: Table 1 & Table 2: Authors should have the kinetic curves of these two substrates and different mutants in supplementary figures so that others can check the fitting of kinetic curves.

Response 1: Based on the reviewer’s comments, we have added the kinetic curves of CSEs with β-CA and cystine metabolism in supplementary figures (see Supplementary Figure 1A and 1B). We made a statement in this regard (lanes 100-103, highlighted in red). Since the standard error of the mean of C136V/C171V was is quite a lot larger than those of other mutants in the original table 2, we did re-tested and re-evaluated the kinetic data of this mutant (see Table 2, lanes 291-300, highlighted in red).  

Point 2: The method part has the pyruvate quantification by coupling assay. However, the main text doesn’t describe this method at all. It should be deleted.

Response 2: We did add this method in the main text, since this method was employed in Table 1. We make a statement in the legend in Table 1 (lanes 301-304, highlighted in red). 

Round 2

Reviewer 3 Report

All concerns have been resolved properly after the major revision of the manuscript.